# Topical Treatments for Basal Cell Carcinoma and Actinic Keratosis in the United States

**DOI:** 10.3390/cancers15153927

**Published:** 2023-08-02

**Authors:** Isabella J. Tan, Gaurav N. Pathak, Frederick H. Silver

**Affiliations:** Department of Pathology and Laboratory Medicine, Rutgers Robert Wood Johnson Medical School, Rutgers, The State University of New Jersey, New Brunswick, NJ 08854, USA

**Keywords:** basal cell carcinoma, BCC, actinic keratosis, AK, topical imiquimod, topical fluorouracil, tirbanibulin ointment, BCC treatment

## Abstract

**Simple Summary:**

Basal cell carcinoma and actinic keratosis are two of the most common cutaneous lesions identified in the dermatology clinic. There is established evidence suggesting that topical treatments play a significant role in treating early forms of superficial BCC while also lowering the economic burden of healthcare by alleviating the need for unnecessary biopsies. Considering the expected continued rise in prevalence of BCC and AK in the coming years, topical therapies can decrease the cost of treatment, limit in-office procedures, and lessen the risks associated with surgery, including infection and scarring.

**Abstract:**

Skin cancer is an overarching label used to classify a variety of cutaneous malignancies. Surgical excision procedures are the commonly used treatments for these lesions; however, the choice to perform operative intervention may be influenced by other factors. Established research and literature suggest that topical treatments limit the need for surgical intervention and its commonly associated adverse effects, including infection and scarring. In addition, the growing indications for the usage of topical therapies in BCC treatment, as well as their increased availability and therapeutic options, allow for their greater applicability in the dermatology clinic. Certain topical therapies have been highlighted in research, especially those targeting basal cell carcinoma (BCC) and actinic keratosis (AK). There is also a clear correlation between cost and treatment outcomes, considering BCC’s ever-growing prevalence and the proportion of excised lesions being reported as malignant. This review will discuss BCC and AK lesion criteria that result in the most successful outcomes using topical treatments, then highlight the various topical treatment options, and finally address their clinical significance moving forward.

## 1. Introduction

Basal cell carcinoma (BCC) is the most common form of skin cancer in the United States, with an estimated 5.2 million cases in the United States alone [1]. It has been shown that the global incidence of BCC is predicted to continue its growth throughout the next 30 years and, according to current projections, will continue to grow at a rate of 3.6 million new cases annually in the United States alone [2]. Due to the slower growth and relatively lower potential for metastasis of BCC, there is frequent underreporting [3]. There is also a probable underestimation of these statistics. Countries with poor cancer reporting registries are likely to underestimate these figures, including sub-Saharan African countries [2]. While there is no direct evidence of geographic regions having a significant difference in preference for topical treatments, certain countries with a lack of healthcare practitioners experience with excisional therapy options would likely exhibit higher rates of topical treatment usage [4]. Considering BCC and AK’s projected rising prevalence in the coming years, there is a significant need to address other treatment modalities, which include topical treatments.

BCC pathogenesis involves a DNA damage trigger that promotes cellular changes in the basal cells of the epidermis, often leading to uncontrolled cell growth [1]. BCC is characterized by dysregulated activation of the Hedgehog signaling pathway due to mutations in PTCH1 or SMO genes, leading to uncontrolled cell proliferation and tumor formation, making the pathway a target for BCC treatments [2]. BCCs are slow growing, which indicates that most are treatable and do not cause excessive damage if identified and treated early in their progression. There is recent research and literature on how best to enhance early diagnosis in hopes of providing early and targeted treatment of the lesion while also preventing remission. BCC can vary in its presentation, ranging from a scaly lesion to shiny papule [2]. In patients with darker skin tones, BCC can take on different features and often present as a pigmented lesion [3]. If left untreated, BCC can become pervasive and invade local tissue, having the potential to destroy deeper tissues and bone.

AK is a precancerous cutaneous growth commonly elicited by chronic sun exposure [4]. AK is one of the most common conditions that dermatologists treat, with an estimated 40 million Americans developing a new AK every year, which is anticipated to continue trending in this direction in the coming years [4]. AK often presents as a scaly lesion on sun-damaged skin and is associated with poor immune function, the use of certain drugs, and an age-related consequence. Like BCC, AK can present in a variety of ways, including a thickened plaque, a white scaly lesion, or a pigmented papule. AK can also present differently on pigmented skin. It was also identified that the number of AK lesions may correlate with the risk of SCC [5,6]. Treatment options for BCC include surgical excision, cryosurgery, radiation therapy, photodynamic treatments, and topical therapies [3]. The decision about which treatment option to pursue typically factors in patient history and age [2]. Treatment options for AK are similar and often include cryosurgery or topical treatments [6]. There is also a significant public health and economic concern, especially considering BCC’s high prevalence, as surgical excision remains the mainstay of treatment options chosen by clinicians. There is also strong significance in considering the clinical presentation and distinguishing between nodular and superficial BCC in making the clinical determination of the treatment approach.

There is a proposed genetic mechanism underlying the development of BCC and AK that suggests an interplay between germline and somatic mutations, specifically those that are inherited variations and somatic mutations in key genes. AK development is associated with p53 gene mutations, which impair DNA repair and increase the risk of malignant transformation [4]. BCC, the most common skin cancer, is primarily driven by mutations in the PTCH1 or SMO genes, resulting in uncontrolled activation of the Hedgehog pathway and thus promoting tumor formation [2]. By developing a better understanding of the genetic factors underlying BCC and AK, there is potential for the development of future targeted treatments.

There is also a cost difference that exists within topical treatments, and studies have shown that 5-FU is more cost-effective than imiquimod topical therapy [7]. In high-risk carcinomas where topical treatments are not suitable, surgical margins are necessary, and if there is local invasion or a lack of wide margins on the face, Mohs surgery is recommended over topical therapies [8]. According to one systematic review, delay in diagnosis was the main determinant of treatment cost, surgical complexity, and procedure type [9]. Size also affects the cost per treatment for nonsurgical options [10]. Data supports that early detection often has better outcomes and decreased average treatment costs, and recent literature suggests that if the lesion is 0.6 cm or less, the treatment cost is decreased [8].

However, as lesion sizes increase, excision becomes comparatively more expensive [9]. Of note, if repair of office-based excisional defects is delayed until negative margins are confirmed by permanent section, the cost increases by about 16% over immediate repair since the multiple surgery reduction does not apply to excision and repair codes on different dates of service [11].

Another significant motivation for choosing topical treatments as opposed to surgical intervention is due to the fact that around 44.5% of cutaneous biopsies performed by a dermatologist are benign [12]. In certain indications, topical treatments for BCC may be used, which can lessen the healthcare burden associated with continuous excision [13]. These may also be augmented by improving the diagnostic tools available for BCC and AK. Considering the recent developments of novel technologies aiming to promote noninvasive screening methods, there will be a shift toward earlier diagnosis of lower-grade lesions, for which treatment modalities such as topical therapies are indicated. Topical treatments can be advantageous to the patient by decreasing the cost of treatment, limiting in-office procedure costs, and lessening the risks associated with excisional biopsy, including infection and scarring, which can positively impact the patient’s quality of life [14,15]. Although there are special considerations that need to be accounted for in the use of topical therapies, such as adverse skin reactions and patient adherence, the benefits may prove to outweigh the drawbacks. Non-invasive tools also have the capacity to potentially augment earlier diagnosis of BCC and AK, which can have crucial implications for the future use of topical therapeutics.

Overall, there is a considerable potential benefit to topical treatments for BCC and AK, and further studies will be crucial in elucidating the role that these therapies play in treating these lesions in the future as BCC and AK prevalence continues to rise.

## 2. Materials and Methods

A literature search was performed using a comprehensive PubMed search to evaluate BCC and AK lesion criteria and topical treatment indications. The search was limited to articles in the English language. All types of studies were included in the literature search. Articles not pertaining to BCC and AK lesion criteria or topical treatment options for BCC and AK were excluded. Titles and abstracts were screened for inclusion and exclusion criteria. The screened papers were then reviewed in full text, and those not found to meet inclusion and exclusion criteria were excluded.

## 3. Discussion

BCC Lesion Criteria

BCC is the most common form of skin cancer [8]. It is a malignancy that manifests in the deepest layer of the epidermis and typically occurs on sun-damaged regions of the skin, including the face, neck, and trunk [3]. BCC has a low mortality and metastatic rate, and its onset is the result of patient exposure to ultraviolet radiation and genetics, among other environmental factors [3]. Clinical features and histopathology are varied depending on the subtype, and dermoscopy is a method used to augment diagnosis [3]. In vivo diagnostic tools are currently being developed that can promote early diagnosis and use less invasive tools, which both enhance the need for greater therapeutic options.

Certain topical treatments are FDA-approved for use in cases of superficial BCC and are formulated as gels or creams to be directly applied to the impacted regions of the skin. These topical therapies treat superficial BCCs with a lower risk of scarring [9]. Less invasive treatments are also being developed, all with the ultimate goal of full resolution, positive cosmetic results, and limited side effects [2].

BCC is often diagnosed by biopsy, and although there are 26 different subtypes, BCC is usually split into three categories: superficial, nodular, and infiltrative [2]. However, considering the overlap between the various subtypes, they are often difficult to discern, even under dermoscopy [3]. The histopathology of BCC includes basal cell aggregates with large hyperchromatic nuclei and smaller cytoplasms encased in a fibrous stroma [2]. Histopathologic characteristics also inform BCC staging, in which subtypes are categorized according to the risk of recurrence. BCCs with a lower risk of recurrence include superficial, pigmented, and nodular BCCs [1]. BCCs with an elevated risk of recurrence include infiltrative, micronodular, and sclerosing BCCs [1].

Although growth is usually localized, BCC can infiltrate local tissue and, if left untreated, cause disfigurement and tissue necrosis, which can be especially problematic in the facial region [2]. Patients typically present with a non-healing, nontender lesion, and many diagnoses are made incidentally, and at a more severe stage [2]. These findings indicate a strong need for enhanced diagnostic tools to augment early diagnosis, which can direct therapeutic treatment and improve clinical outcomes.

Early diagnosis is also paramount considering that small BCCs are often difficult to distinguish with the naked eye and under dermoscopy [2]. Novel technologies, including optical coherence tomography, are currently being developed and tested in order to augment early diagnosis in hopes of improving clinical outcomes and limiting unnecessary surgery and biopsy [8]. Considering the pattern that early BCCs are easily missed, novel technologies that are non-invasive, effective at diagnosing early lesions, and seamless to use would greatly improve clinical outcomes. Topical treatments play a key role in this relationship, as they are often indicated for lower-grade BCCs [8]. As methods of early diagnosis continue to be developed and improved upon, there will be an appreciable rise in the need for and market for topical therapies targeting BCC.

### 3.1. Superficial

The superficial BCC presents as a thin, pink, irregularly shaped lesion surrounded by a pearl-like border [8]. Under histopathological examination, these are identified by multiple basaloid cell nests seen in the epidermis, with no deeper dermal invasion [8]. This subtype of BCC is most commonly seen in younger age groups [8]. It commonly arises in sun-exposed regions, most frequently on the upper trunk and shoulder areas. The diameter may vary, often between a few millimeters and a few centimeters. Secondary features may include a focal crust, a rolled border, and variable levels of melanin [10]. In larger lesions, hypopigmentation and atrophy may be present as a result of spontaneous regression [10]. It is often seen in a multifocal pattern, which may lead to incomplete excision if surgery is performed [10].

### 3.2. Nodular

Nodular BCC is the most common type of BCC found on the head and facial region. It typically presents as a pink papule with small blood vessels, or telangiectasias, covering the surface of the lesion [8]. It often has a shiny or pearl-like surface and may have a central depression, which can give the lesion a rolled appearance [10]. Under histopathology, there are cell aggregates with sporadic arrangements of central cells [8]. Nodular BCC also has multiple subtypes according to certain secondary features, including keratotic (having mature keratin central spots) and cystic (having cystic degeneration) [10]. This subtype may also form an ulcer in larger lesions in which a sharp border is still present, which is often a significant clue in diagnosis [8].

### 3.3. Infiltrative

Infiltrative BCC is less common but has a more distinct and severe presentation, which often includes local tissue invasion [8]. On gross examination, it can appear as a scar-like growth, be shiny, or exhibit telangiectasias [8]. It can present histologically as thick strands that can take on a spiky and irregular appearance [10]. It is most commonly seen in the face and trunk and typically presents as lightly colored and ill-defined due to its development and invasion between collagen fibers. Infiltrative BCC also forms in small clusters, which can limit its identification and treatment [10].

### 3.4. BCC Topical Treatments (Table 1)

Clinical Reasoning/Indication for Topical Treatment [16,17,18,19,20,21,22,23,24,25,26,27,28,29,30,31,32,33,34,35,36,37,38,39,40,41,42,43,44,45,46,47,48,49,50]

According to its clinical features, BCC can be classified into low- and high-grade lesions. Clinical factors include lesion size, margins, location, recurrence, and histological features. Locations that are commonly indicated as intermediate and higher risk include the nose, head, and neck [11]. Lower-risk locations often include the trunk and extremities [11]. This clinical determination is necessary in order to direct the treatment approach, as low-grade lesions are indicated for topical treatment [11].

**Table 1 cancers-15-03927-t001:** Topical treatment options for BCC.

Topical Treatment	Description	Indication
Imiquimod cream [16]	Stimulates immune response against BCC cells	Superficial or low-risk BCCs
5-fluorouracil (5-FU) [25]	Inhibits cell growth and promotes cell death	Superficial or low-risk BCCs
Diclofenac gel [30]	Nonsteroidal anti- inflammatory medication	Superficial or low-risk BCCs
Ingenol mebutate gel [50]	Induces cell death and local immune Response	Superficial or low-risk BCCs

In early-stage lesions, BCC subtypes are often difficult to distinguish, which limits their diagnosis with the naked eye and dermoscopic tools. Novel technologies, including optical coherence tomography, are currently being developed to augment early diagnosis, which can have a crucial role in topical treatment indication and improving overall clinical outcomes [11].

Topical treatments are generally recommended as a second-line treatment option for superficial basal cell carcinomas (sBCC). There is limited evidence of efficacy in nodular subtypes, but topical treatments are indicated for use in low risk, superficial, or nodular lesions. Features associated with lower risk are lesions <1 cm in diameter on the trunk and extremities, which are nodular or superficial growth patterns that lack perineural invasion [11]. sBCCs comprise 15–30% of all basal cell carcinomas and generally occur more frequently in females, at a younger age, and in the trunk region [12]. Superficial BCCs are also more common in patients who have a previous medical history of basal cell carcinoma. Although surgical excision has the lowest 5-year recurrence rate of any treatment option, imiquimod (remission rate of 82–90%) and topical 5-fluorouracil (5-FU) (80%) are both effective treatment options when surgical treatment is not preferred [13].

Topical agents are efficacious in treating sBCC and have a variety of advantages over surgical treatments. Studies have suggested that patients prefer topical medications over surgery for BCC treatment due to fewer side effects, cosmetic implications, and lower costs [15]. Topical treatments may also be more advantageous in patients with multiple, low-risk BCC lesions where multiple excisions would be undesirable [16,17,51]. One 5-year clinical study evaluated the efficacy of imiquimod and topical 5-FU in sBCC and concluded that imiquimod was superior to 5-FU and photodynamic therapy in preventing sBCCC recurrence [18]. Given that sBCC recurrence is common in patients with previous BCC, this may help when topical therapy is preferred.

Given the rise in development and usage of non-invasive diagnostic methods, BCC lesions are being recognized in earlier stages, which informs the future use and clinical applicability of topical treatments, for which these treatments are indicated.

There are concerns surrounding topical treatment use for BCC. Currently, these treatments are indicated for low-grade lesions, and even in a clinically indicated topical approach, there are challenges of patient self-application, potential side effects, and the necessity of dermatologist supervision.

Imiquimod 5% Cream

Imiquimod cream is FDA-approved for primary superficial BCC in immunocompetent adults with a maximum lesion diameter of 2 cm and a 1 cm margin (3 cm total) located on the trunk, neck, or extremities when surgical methods are less appropriate [19]. The typical treatment regimen is imiquimod 5% cream at bedtime, five days a week, for six weeks. The mechanism of action of imiquimod is by directly binding to toll-like receptors (TLR) and inducing the release of inflammatory immunomodulatory cytokines [20].

Although not clinically indicated, imiquimod has been used successfully for nodular BCC (70% clearance rate) [21]. The immune modulating effect of imiquimod may produce longer clinical efficacy compared to 5-FU treatment with a lower BCC recurrence rate in 5-year follow-up studies [22]. The most commonly reported adverse effects of imiquimod are local skin reactions such as burning, pain, and itching, which are similar to 5-fluorouracil.

5-fluorouracil (5-FU) Cream

Topical 5-fluorouracil 5% cream is FDA-approved for the treatment of superficial BCC when other treatment options are impractical. This medication is a pyrimidine antimetabolite that interferes with DNA synthesis, inhibits cell proliferation, and causes cell death. It is typically administered twice daily for 3–6 weeks [23].

Studies have shown that topical 5-FU had a low rate of failure compared to the surgical treatment group and the destructive management (cryotherapy, etc.) group for SCC in situ and superficial BCC [24,25]. It has been shown to be effective in the treatment of superficial BCCs [26].

### 3.5. AK Lesion Criteria

Actinic keratosis (AK) is also known as solar keratosis and typically presents as a patch of rough, scaly skin [4]. It is a precancerous lesion that is histopathologically identified by atypical keratinocytes in the basal layer of the epidermis. Under dermoscopy, AKs can be identified by red or brown networks, surface scales, gray dots, and circular structures [27]. Using newer technology, such as in vivo microscopy, AKs may present as a disarranged honeycomb pattern [27]. Typically, clinical features include erythematous macules or papules, which can be pigmented or ulcerated [4]. These typically also coincide with a patient’s history of chronic sun exposure or sun-damaged skin. However, there are various possible presentations of AK, including a wart-like surface, crusting, burning, or color variations [27]. It is closely associated with a history of sun exposure and is typically found on the forearms, face, ears, and scalp. Due to its close association with sun exposure, AK is typically slow-growing and often presents in patients over 40 years of age [28]. If left untreated, there is a risk of AK progression to squamous cell carcinoma (SCC) [5].

The high prevalence of AK indicates an elevated burden of disease. Considering the higher prevalence in middle-aged and elderly populations, along with the recent pattern of the aging population, there is a strong push for the development of novel technologies to identify AK lesions early in their progression, for which topical treatments are indicated.

Classification and staging of AK are difficult as there are varied opinions on how to account for specific clinical, dermatoscopic, and histological features of the lesion. The variations in presentation of AK preclude early diagnosis with the naked eye and under dermoscopy; however, recent technological advances can improve and augment early diagnosis. By improving these metrics, there will be a greater need for topical treatment usage, which can limit the need for biopsy and surgery and the costs and complications associated with those.

One grading scheme deemed that the visible and clinical features of AK were paramount, leading to the development of three grades to assess severity [27]. However, other methods include solely relying on histopathological examination to classify into the grading system [28]. According to the histological approach, histological examination reveals multiple AK subtypes, including hypertrophic, atrophic, lichenoid, and pigmented [28]. In either method, classification of early-stage AK is difficult and often precludes accurate diagnosis and treatment [27]. Overall, this pattern reflects that the level of agreement between the clinical and histological grading scales is low, indicating the importance of early diagnosis and treatment for all AK lesions, regardless of stage and severity. With the development of novel technologies to augment earlier and more accurate diagnosis, there will be a greater market need for treatments that target lower-grade lesions.

Treatment goals for AK include removal of the clinical lesion, prevention of evolution to SCC, and a reduction in AK relapse. When considering different treatment options, there are multiple approaches, which often include lesion-directed and field-directed therapies. Lesion-directed therapies include surgical intervention, cryotherapy, and laser therapy [29]. Field-directed approaches include topical treatments as well as photodynamic therapy [30]. The main advantage of topical treatments is their ability to treat multiple AK lesions in one area at once. This approach of using topical medications can also prevent new AK formation and cutaneous malignancies in the future [31]. The downsides of topical therapies include adverse skin reactions as well as the requirement of strict medication adherence [31]. In order to be effective, the topical treatment must be applied as often as the prescriber recommends, even if an adverse skin reaction occurs and persists [31]. However, once the topical treatment regimen is complete, new, healthy skin will be produced.

There are certain AK lesion criteria that are associated with topical treatment indications, which include “field cancerization”, which is characterized by multiple AK lesions and chronic cutaneous damage in adjacent areas. Damage associated with field cancerization is classified by having at least two of the following characteristics: dyspigmentation, atrophy, rough texture, and telangiectasias [30]. Patients fulfilling the characteristics of field cancerization have an increased risk of progression to SCC and are best treated using field-directed therapies [32]. Field-directed treatments for AK include topical therapies such as 5-fluorouracil (5-FU) cream, imiquimod cream, tirbanibulin ointment, and diclofenac sodium gel, each with unique characteristics, adverse reactions, and indications [30].

AK recognition and diagnosis are often based on clinical appearance and patient history; however, recently, technologies have been focused on early diagnosis, especially those utilizing reflectance confocal microscopy and optical coherence tomography, which may allow for the detection of clinically invisible lesions and also provide a method for easy treatment efficacy monitoring [31]. As in BCC, AK lesions have been the target of recent scientific innovations, especially as they relate to non-invasive diagnostic methods to augment early diagnosis [31]. These developments direct earlier diagnosis and intervention, which can potentially reduce the need for unnecessary surgery, biopsy, and their associated complications. Topical treatments have been of recent interest in the treatment of early-stage AK, and with the rise of early-stage lesion identification, there will be a significant need for AK topical treatment usage and indication in the dermatology clinic.

AK Topical Treatments

Like sBCC, actinic keratosis can be managed with 5-FU and imiquimod topical creams (Table 2). For patients with multiple thin lesions on the face or scalp, 5-FU topical treatment is recommended as a first-line therapy [5,33]. Patients with multiple AKs (including hyperkeratotic lesions) may benefit from a combination of topical and invasive treatments. Imiquimod 2.5%, 3.75%, and 5% creams are FDA-approved for the treatment of typical hyperkeratosis visible on the full face or scalp in immunocompetent adults. The mechanisms of action of these drugs are similar to those of BCC. Treatment with imiquimod has been shown to be effective, with only 10% of patients developing subsequent AKs in the treatment field within 12 months [34,35]. Another comparator study showed that imiquimod treatment of AK resulted in superior sustained clearance and cosmetic outcomes compared to cryosurgery and 5-FU [36].

Tirbanibulin 1% ointment

Tirbanibulin 1% ointment is FDA-approved for the treatment of AK occurring on the face or scalp. It is efficacious in AK, with 49% of patients reporting complete resolution of AK and 72% of patients reporting at least partial clearance. The median reduction in lesion count is 87.5% [37]. It is well tolerated, with mostly mild transient application-type reactions (pruritus) and limited phototoxic skin or sensitization reactions [38,39].

Diclofenac 3% non-steroidal anti-inflammatory sodium gel

Diclofenac is a 3% non-steroidal anti-inflammatory sodium gel agent that is an FDA approved agent for AK. It treats AK by inhibiting COX 2, reducing angiogenesis, and cellular proliferation. Clinical studies have shown that four weeks after 60 days of treatment, 33% of patients had a complete response, and after 90 days of treatment, 50% of patients had a complete response to treatment [30].

## 4. Conclusions

BCC and AK are cutaneous lesions that are typically treated using surgical methods [40,41]. These methods are often associated with greater costs to the patient as well as a greater risk of complications, such as infection and scarring. However, there is recent evidence that topical treatment modalities are increasing in efficacy and should be directed for greater use in the dermatology clinic in hopes of avoiding these potentially undesirable effects [42,43,44]. Topical therapies are often indicated for AK and superficial BCC, and each has its own distinct characteristics, indications, and limitations. However, there is clear evidence suggesting that topical treatments may have a significant role in treating early forms of BCC and AK while lessening the economic healthcare burden by alleviating the need for unnecessary biopsy and surgical interventions [45].

With the rise in the development of novel diagnostic tools allowing for the earlier recognition and diagnosis of BCC and AK, there is a growing need for therapies targeting smaller and earlier-stage lesions [46]. Topical treatments have been shown to be an effective tool for targeting these early lesions and thus have the potential to play a key role in the future treatment approach for BCC and AK in the dermatology clinic [47]. Also, considering the anticipated continued rise in prevalence of BCC and AK in the coming years, topical therapies have great potential for decreasing the overall cost of therapy, limiting in-office procedures, and lessening the risks associated with biopsy and surgery [48,49].

Further studies should be conducted to elucidate these relationships; however, the need for more widespread use of topical treatment modalities for BCC and AK in the dermatology clinic is clear in order to improve future clinical outcomes and avoid unnecessary surgery and its associated complications. However, currently, surgery remains the “gold standard” for treatment for BCC and AK.

## Figures and Tables

**Table 2 cancers-15-03927-t002:** Topical treatment options for AK.

Topical Treatment	Description	Common Side Effects
5-Fluorouracil (5-FU) [31]	Inhibits DNA synthesis and cell proliferation	Local skin reactions, erythema, skin peeling, itching
Imiquimod [31]	Induces local immune response	Local skin reactions, redness, itching, burning sensation
Ingenol mebutate [52]	Induces cell death and inflammation	Local skin reactions, erythema, crusting, swelling
Diclofenac [30]	Anti-inflammatory and immune- modulating	Local skin reactions, redness, itching, rash, dryness
Tirbanibulin [37]	Disrupts microtubule network and inhibits cell division	Local skin reactions, erythema, pain or burning sensation

## Data Availability

References used in this paper available by writing to silverfr@rutgers.edu.

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
