# Peer review of "Topical Treatments for Basal Cell Carcinoma and Actinic Keratosis in the United States"

_cancers, 2023, doi:10.3390/cancers15153927_

Round 1
Reviewer 1 Report (Previous Reviewer 1)
Dear authors,
Firstly, I would like to congratulate you and your effort in writing a revision article.
In the abstract you are talking about recent topical treatments, twice, and that is a mistake as Imiquimod or 5-Fluorouracil, for example, have been being used for more than a decade.
Introduction
Second paragraph, pathogenesis of BCC: you should mention the hedgehog molecular pathway.
Third paragraph:
· The risk of evolution of an AK to an SCC is not such high are you are describing and referring to reference 6.
· Metastasis status in a decision of manage a BCC is exceptional as metastasis are completely infrequent.
· Treatment of AL with surgical excision is unnecessary and not indicated.
· The description of the genetic underlying mechanism under BCC and AK are quite simple, in AK p53 mutation are very well known and crucial.
· Surgical margins are needed in high-risk carcinomas in which topical treatments are contraindicated, in case of local aggressivity and lack of wide margins in the face, Mohs surgery is indicated, but not topical treatments.
· The recognition of the clinical…you are repeating the idea of the previous paragraph.
· Revise that reference, and check, it seems to me a high percentage of benign biopsies when malignancy is suspected. That is not an argument for using topical treatment, as it depends on the type of BCC not on economic arguments. A biopsy of a small BCC sometime is used instead for complete excision in once.
· Last paragraph, maybe you could add some Guidelines which describes the role and when to use every topical treatment that you are describing, you could check de guidelines of the American Academy of Dermatology in the management of BCC and AK.
The rest of the manuscript is complete correct and well written. Maybe it is too general, I mean you can go depth in any point as you are talking about many different options for the treatment of two different pathologies. The discussion is superficial does not add any new.
English language is correct
Author Response
see attached

Reviewer 2 Report (Previous Reviewer 2)
I m agree for pubblication.
I already suggest to add some clinical photo og parients before and after treatments.
None
Round 2
Reviewer 1 Report (Previous Reviewer 1)
Dear authors,
I believe you have improved your manuscript.
In the conclusion, when you say: “Lessening the risk associated with biopsy and surgery”. This is not completely correct, as many Basal cell carcinomas and actinic keratosis need to be biopsied to confirm the diagnosis and the histological pattern. After, the treatment with topical treatment is decided if it is possible. For all that reason I believe you should eliminate biopsy, as is the gold standard diagnosis method, and left only surgery.
Best,
Author Response
see attached

This manuscript is a resubmission of an earlier submission. The following is a list of the peer review reports and author responses from that submission.
Round 1
Reviewer 1 Report
Dear Authors,
I fell that your manuscript it Is not well written, half of the manuscript is a description of BCC and AK with a low level. I mean this is not interesting for a dermatologist, whom I believe would be the main readers.
On the other hand, the description of the treatments is superficial and very simple, you do not add a Table, or a description of clinical trials.
I am sorry but form my point of view it is not suitable for publication in Cancers.
Here are my comments, if you want to check and of course, improve it. You should continue trying.
Line 12: I would say not high risk carcinoma,, in non risk areas, that it is important in the face
Line 69: I believe it is important the clinical presentation, if you are suspecting a nodular BCC it is not the same as a superficial.
Also it is very important to consider what it is definided as Hifh risk basal cell carcinoma ( Check for example the last guides of the British Journal of Dermatology. It those cases, topical treatments are not indicated.
Line 74: Mohs excision it is not comparable, there is a mistake here from my view. Mohs excision it is neccesary in selected cases of BCC. That it is not an argument
Line 90: Again the main decission for choice any or other treatment relies on the clinical exploration or of course, in the gold standard, a biopsy
Line 94: Please review the literature, dermatologist are most trained doctor in images classification
Line 111: You should explain de no benefits, as patient application at home, time consuming with crustering or side effects, uncontroled by the dermatologist and patients´satisfaction
Line 131: This paragraph is redundant
Line 152: Please try to summarize, this information is redundant
Line 194: Again those parragraphs are redundant and inconcluse, please try to sum up or ad a table describing the type of BCC.
Line 238: You are in page 6 and not having talk about therapies yet
Line 267: The revision it is superficial, nothing is added to a dermatologist
Best,
Dear Editor,
I believe English it is ok, but not the manuscript.
Reviewer 2 Report
dear authors, NMSCs represent a very important class of cancers linked to a number of risk factors. I read this review with great interest and I have some suggestions to offer: In the title I would indicate the country of Reference, given that in the text you refer several times to the USA. I would underline in the introduction the genetic link of the onset of BCC and AK doi: 10.1684/ejd.2021.4092.
For BCC: Why do you describe only two treatments?
SEe doi: 10.3390/biomedicines8060156.
References should be increased.